# Antibody Fragments as Tools for Elucidating Structure-Toxicity Relationships and for Diagnostic/Therapeutic Targeting of Neurotoxic Amyloid Oligomers

**DOI:** 10.3390/ijms21238920

**Published:** 2020-11-24

**Authors:** André L. B. Bitencourt, Raquel M. Campos, Erika N. Cline, William L. Klein, Adriano Sebollela

**Affiliations:** 1Department of Biochemistry and Immunology, Ribeirao Preto Medical School, University of São Paulo, Ribeirão Preto, SP 14049-900, Brazil; brandaobqi@hotmail.com (A.L.B.B.); raquelmariacampos@usp.br (R.M.C.); 2Department of Neurobiology, Northwestern University, Evanston, IL 60208-3520, USA; erika.cline@northwestern.edu (E.N.C.); wklein@northwestern.edu (W.L.K.)

**Keywords:** antibody fragments, single chain, amyloid, oligomer, neurotoxicity, NUsc1

## Abstract

The accumulation of amyloid protein aggregates in tissues is the basis for the onset of diseases known as amyloidoses. Intriguingly, many amyloidoses impact the central nervous system (CNS) and usually are devastating diseases. It is increasingly apparent that neurotoxic soluble oligomers formed by amyloidogenic proteins are the primary molecular drivers of these diseases, making them lucrative diagnostic and therapeutic targets. One promising diagnostic/therapeutic strategy has been the development of antibody fragments against amyloid oligomers. Antibody fragments, such as fragment antigen-binding (Fab), scFv (single chain variable fragments), and VHH (heavy chain variable domain or single-domain antibodies) are an alternative to full-length IgGs as diagnostics and therapeutics for a variety of diseases, mainly because of their increased tissue penetration (lower MW compared to IgG), decreased inflammatory potential (lack of Fc domain), and facile production (low structural complexity). Furthermore, through the use of in vitro-based ligand selection, it has been possible to identify antibody fragments presenting marked conformational selectivity. In this review, we summarize significant reports on antibody fragments selective for oligomers associated with prevalent CNS amyloidoses. We discuss promising results obtained using antibody fragments as both diagnostic and therapeutic agents against these diseases. In addition, the use of antibody fragments, particularly scFv and VHH, in the isolation of unique oligomeric assemblies is discussed as a strategy to unravel conformational moieties responsible for neurotoxicity. We envision that advances in this field may lead to the development of novel oligomer-selective antibody fragments with superior selectivity and, hopefully, good clinical outcomes.

## 1. Toxic Protein Oligomers in Central Nervous System Diseases

In living systems, proteins must assume and maintain a three-dimensional conformation, which dictates their biological functions. Under certain conditions, however, monomeric protein units may self-associate to form oligomeric structures that display both loss of biological, and gain of toxic, function [1]. Ultimately, these oligomers have the potential to aggregate into insoluble amyloid fibrils, highly stable non-branched insoluble structures rich in β-sheet content [2,3,4]. Although this property is inherent to all proteins [5,6,7], a number of amyloidogenic proteins accumulate in tissues, causing diseases known as amyloidoses, which can be systemic but commonly impact the central nervous system (CNS) [1,8,9,10,11]. 

It is now evident that soluble oligomers are the most toxic form of amyloidogenic proteins, more so than their monomeric or fibrillar forms, disrupting, e.g., synaptic function, membrane permeability, calcium homeostasis, gene transcription, mitochondrial activity, autophagy, and/or endosomal transport in an array of disease models [12,13,14,15]. The first reports on the brain accumulation of toxic soluble oligomers were in Alzheimer’s disease (AD); the associated oligomers mainly composed of the 4.5 kDa amyloid β (Aβ) peptide [16,17,18]. Since then, toxic soluble oligomers of other proteins have been implicated in the onset and progression of several debilitating CNS diseases, e.g., tau, α-synuclein, the prion protein (PrP^c^), and huntingtin protein (htt) in Alzheimer’s, Parkinson’s, prion, and Huntington’s diseases, respectively [19,20,21,22,23,24]. In fact, many of these protein oligomers are found together in multiple diseases [25,26]. 

Amyloidogenic oligomers have been frequently implicated as promising diagnostic and therapeutic targets for CNS amyloidoses [12,14,27,28,29,30,31,32,33]. Despite their disease relevance, the structural hallmarks of such soluble oligomers remain elusive due to their metastability and heterogeneity, hampering our ability to target them therapeutically and diagnostically [12,34,35,36]. One promising strategy in the structural analysis of amyloidogenic oligomers is the utilization of antibody fragments, which can achieve high conformational selectivity, enabling the isolation and stabilization of different oligomeric species. Furthermore, the structural properties of the antibody fragments themselves make them promising diagnostic/therapeutic tools. In this review, we discuss their application as tools for structural research and diagnostic/therapeutic targeting of oligomers acting in brain amyloidoses.

## 2. Antibody Fragments

Monoclonal antibodies (mAbs) are currently the largest, and most rapidly growing, class of biopharmaceuticals on the market to treat a variety of diseases [37,38]. However, only four mAbs have been approved to treat a neurodegenerative disease (multiple sclerosis), and these antibodies are thought to work primarily in the periphery [37]. There are a number of challenges in utilizing monoclonal antibodies for the diagnosis or treatment of brain diseases. For one, their large molecular mass hinders their ability to cross the blood–brain barrier [38,39]. Moreover, the crystallizable fraction (Fc) of mAbs can mediate deleterious inflammatory responses resulting in, e.g., meningoencephalitis, vasogenic edema, cerebral microhemorragies, and even death [40,41,42,43,44,45,46,47]. Regarding diagnostics, poor contrast of mAbs in imaging applications due to a long serum half-life has been reported as a drawback [48]. 

During the past 20 years, antibody fragments have been developed as an alternative to full-length IgGs for the diagnosis and treatment of a variety of diseases, including brain disorders [13,38,39,47,49,50,51,52]. These molecules are simple protein motifs of large diversity that include the IgG antigen-binding domain(s) but lack the inflammatory Fc domain, retaining the total (fragment antigen-binding: Fab and single-chain variable fragment: scFv) or partial (VH) antigen specificity of intact IgGs [38,39,52]. 

Compared to full-length IgGs, antibody fragments have advantages and disadvantages as therapeutics. An important advantage is their smaller size (12–50 kDa), thought to potentiate the blood–brain barrier crossing and tissue penetration and enable access to challenging, cryptic epitopes [38,39,52]. Furthermore, their fast blood clearance makes them ideal imaging agents [39]. On the other hand, their smaller size leads to a shorter half-life in vivo, in part due to rapid kidney clearance, which limits the chance of target engagement without the addition of half-life extension moieties (e.g., PEG and albumin-binding fragments) [38,44]. Another advantage of antibody fragments, is their lack of the inflammatory Fc domain (see discussion above). On the other hand, it is noteworthy that the lack of Fc-dependent activation of immune cells may reduce the efficiency of an immunotherapy when a robust inflammatory response is required [53,54], as in cancer immunotherapy, which requires T cell recruiting [55]. 

Another advantage of antibody fragments, is their excellent manufacturability and low cost of production [38,39]. They can be efficiently selected from in vitro display libraries (phage or yeast) and cloned and expressed in heterologous expression systems (e.g., bacteria); this facilitates the production of large quantities in an easy and affordable way. Importantly, the in vitro approach eliminates animal immunization, which may be key when the conformation of the immunogen plays a role in antibody specificity [46,56]. Finally, engineered antibody fragments yielding multimers (diabodies, triabodies, and tetrabodies) have been shown to present higher avidity and lower blood clearance than their monomeric counterparts without compromising tissue penetration abilities [38,48,54]. 

The main types of antibody fragments under development are Fab, scFv (single chain variable fragments), and heavy chain variable domain VH/VHH (single-domain antibodies) fragments [38,39,49,52,57]. The potential of isolated light chain variable domain (VL) chains has not been significantly investigated due to their low stability [56]. An overview of the structures of these molecules is presented in Figure 1. The first artificial antibody fragments reported in the literature were initially obtained by removing the Fc domain through proteolysis [44]. Later advances have enabled the further reduction of antibody structure to scFv and VH/VHH (also called minibodies or nanobodies) [38,39,52,53,54,57]. These fragment types are described in more detail below.

Fab fragments are independent structural units of ~50 kDa containing two antigen-binding sites, with the heavy chain variable domain (VH) linked to the heavy constant domain 1 (CH1) and the light chain variable domain (VL) linked to the light constant domain (CL) [44]. These domains interact through a large interface between the chains (VH/VL and CH1/CL) and a small one between the variable and the constant domains (VH/CH1 and VL/CL) of each chain [59]. The packing between the variable domains creates the antigen binding site [56]. The CH1 and CL domains are also covalently connected by a disulfide bond between Cys residues at their carboxyl terminal region [60,61]. Each Ig domain presents two layers of β-sheet structures, with three to five β-sheets per layer. The variable Ig domains (cyan; Figure 1B) are slightly longer than the constant domains (dark blue; Figure 1B), as they contain two more β-sheets per layer. The β-sheets are connected through loops, and the β-sheet layers of constant domains are attached through a disulfide bond. All amino terminal variable domain loops pack together in a β-sheet motif arranged as an antiparallel barrel-like structure, forming the complete complementarity-determining region (CDR), which is ultimately responsible for the antibody specificity (highlighted in red in Figure 1B) [44,59]. Each Ig domain contains three amino terminal loops encoding different CDR segments. Since the sequence variation associated with the specificity of immunoglobulins is found in CDRs, these regions are also referred to as hypervariable regions [59]. The hypervariable regions assemble into the antigen binding site and interact directly with the epitope. The framework regions, those comprising the variable domain sequences besides CDRs, fold into β-sheet motif structures and provide the scaffold for antibody-antigen interactions [62]. 

Single-chain variable fragments (scFvs), the smallest antibody fragments containing a complete antigen binding site, are recombinant molecules of ~30 kDa in which the variable domains of both VL and VH chains are engineered into a single polypeptide chain connected by a flexible peptide linker and/or a disulfide bond [20,43,45,46]. Their hypervariable segments (amino terminal loops) are approximately 10 amino acid residues long and, as in full length IgGs, form the antigen binding site [59]. The length and amino acid composition of the linker are crucial in maintaining the correct fold of these proteins [54]. The linker is typically about 3.5 nm in length and must contain small, hydrophilic residues (typically Gly and Ser) for enhanced solubility and flexibility [44,54]. 

VH/VHH fragments (~15–20 kDa) are N-terminal Ig domains derived only from the heavy chain, thus retaining antigen binding specificity within a single polypeptide domain [53,59,63]. Similar to VH fragments (Figure 1), VHHs (high affinity variable domains naturally found in camelids) contain three CDRs forming the antigen binding site [59,62]. Human VH domains and camelid VHH framework regions show a high sequence homology [61]. VHH fragments are naturally occurring [38,39,49,52] and especially stable.

## 3. Antibody Fragments Assisting the Study, and Diagnostic/Therapeutic Targeting, of Neurotoxic Amyloid Oligomers in CNS Amyloidoses

In the last two decades, several studies using antibody fragments to study the role of protein oligomers in CNS amyloidoses have been published (Table 1). Considering the discussion in the first two sections above, a major motivation for the use of antibody fragments as research and diagnostic/therapeutic tools for this disease class is the augmented chance of obtaining high affinity, conformation-sensitive antibodies over the typical animal immunization approach. Antibody fragments that display high selectivity for toxic oligomeric conformations are likely to be capable of neutralizing these neurotoxic aggregates without interfering with the physiological function of their monomeric counterparts, therefore presenting as preferred candidates for immunotherapies to treat amyloidogenic diseases. In the following sections, we review studies describing conformational antibody fragments capable of recognizing soluble oligomeric species formed by distinct proteins linked to prevalent CNS amyloidosis that currently lack a cure. We also highlight reports that, in our view, should provide guidance for the development of improved antibody fragments targeting neurotoxic oligomers.

### 3.1. Alzheimer’s Disease

#### 3.1.1. Amyloid β

The increasing collection of antibody fragments against toxic aggregates associated with Alzheimer’s Disease (AD) has enabled the elucidation of important information related to the biochemical nature of these toxic aggregates and their contribution to AD pathogenesis. As discussed above, a major challenge for all amyloidogenic proteins, but perhaps especially for the AD toxins Aβ oligomers (AβOs), has been to identify the most toxic aggregated species. This difficulty in characterization is due to the heterogeneous distribution of metastable species (including non-toxic or differentially toxic species) formed during the aggregation process [81]. Although robust evidence suggests that soluble AβOs and protofibrils play a prominent role in AD progression [12,82], the precise structural features of these soluble aggregates that contribute to AD pathogenesis remain elusive [1,12,81]. However, recent advances in this area have been made possible with the use of conformation-selective fragment antibodies [64,65,66,67,68,69,70,71,72,82,83]. One of those is the scFv antibody NUsc1, selected from a phage-display library by our group [64,65]. NUsc1 presents a marked selectivity for soluble AβOs compared to monomers or fibrils (Figure 2A) and, importantly, provides neuroprotection against AβO toxicity in cell cultures, blocking AβO binding and reducing AβO-induced oxidative stress and Tau hyperphosphorylation [64,65]. NUsc1 is of particular interest since it recognizes a unique conformational epitope displayed on oligomers of Aβ but not those formed by other proteins (such as Tau or Lysozyme); other anti-AβO scFvs have been shown to recognize a common epitope present on oligomers formed by different proteins [73,81,84]. Moreover, NUsc1 exhibits a marked oligomer size-dependent selectivity, preferentially targeting neurotoxic AβO species larger than 50 kDa, as analyzed under non-denaturing conditions by size-exclusion chromatography (Figure 2B). 

Other anti-AβO scFvs have been reported that are promising tools for the study of AβO structure–toxicity relationships as well as their diagnostic and therapeutic targeting. The scFv MO6 was found to target AβO species (18–37 kDa) that are on-pathway to fibril formation and toxic to SH-SY5Y cells [66]. Important to its diagnostic/therapeutic potential, MO6 was demonstrated to cross the blood–brain barrier (BBB) in an in vitro BBB model with a delivery efficiency of 66% 60 min post-administration. Another study reported the scFv b4.4, which recognized an epitope in the central region of Aβ42 (comprising residues H^13^, K^16^ V^18^, F^19^) and was able to neutralize the toxicity of either AβOs or fibrillar Aβ to SH-SY5Y cells [83]. The scFv AS was found to recognize cytotoxic medium-sized AβO species (25–55 kDa) and protofibrils [67]. While scFvs are commonly identified via phage display, AS was identified from a library constructed from the immune repertoire of AD patients. The scFv HT6 also was found to bind efficiently to an N-terminal epitope present in cytotoxic medium-sized AβOs (mainly 18–45 kDa) in vitro [68]. Significantly, the anti-AβO scFv 11A5, selected by phage display and found to target a 34 kDa assembly, has been reported to ameliorate cognitive decline in rats induced by injection of AβOs [69]. It is important to consider that in all of these studies, AβO size has been evaluated by denaturing SDS-PAGE/Western immunoblotting, and therefore may not accurately reflect the AβO size in the physiological milieu. Additionally, an interesting approach has been developed wherein atomic force microscopy is utilized to biopan for conformation-selective antibodies by phage display. Following this approach, two scFvs were identified, named A4 and E1, that targeted distinct oligomeric species presenting either high [70] or low [71] cytotoxicity potentials. Further studies with these conformer-selective scFvs, and others like them, promise to shed additional light on the AβO structural properties contributing to AD pathogenesis.

The scFvs highlighted above were all identified by their unique selectivities from antibody libraries. One promising strategy for the rational engineering of scFvs with even further improved selectivity for oligomeric species of interest, is complementarity-determining region (CDR) mapping (i.e., determination of the complementarity-determining region (CDR) amino acid sequences, the regions responsible for antibody specificity) of existing scFvs. So far, CDR mapping has only been reported for non-conformational anti-Aβ scFvs. In one of these reports, Tiller and colleagues (2017) used a series of mutations in the CDR sequences of scFvs to identify the contribution of arginine residues to the affinity and selectivity for Aβ monomers [85]. Other recent studies have contributed to the identification and importance of particular amino acids within CDRs, e.g., tyrosine, glycine, serine, and especially arginine, in the binding to different Aβ aggregated species [86,87]. If similar studies are conducted with anti-AβO scFvs in the future, comparison to these data obtained with non-conformational anti-Aβ scFvs may indicate the key interactions underlying conformational preference for oligomeric over monomeric and fibrillar species. From a therapeutic perspective, the ectopic expression of neurotoxic-selective fragment antibodies by using brain-optimized viral vectors is emerging as an exciting path to be exploited. For instance, recent data in AD-mouse models indicate a cognitive benefit provided by the brain expression of the scFv NUsc1, which was discussed above (unpublished data [88]).

#### 3.1.2. Tau

Another AD-relevant amyloidogenic protein is the microtubule-associated protein Tau. Upon abnormal hyperphosphorylation or co-factor binding, this protein forms oligomers and larger aggregates that contribute to neuronal dysfunction and death in AD and other tauopathies (reviewed in [89,90]). Since Tau oligomers have been linked to neurodegeneration, structural studies aimed to unravel the conformation of soluble Tau aggregates have been the focus of recent investigations [91]. As with AβOs, antibody fragments are emerging as promising tools for these studies [74,75,92]. For instance, Tian et al. (2015) reported the selection of three conformation-selective anti-Tau scFvs (F9T, D11C, H2A) capable of binding trimeric but not monomeric or fibrillar Tau [74]. These scFvs distinguished AD from cognitively normal post-mortem human brains and are capable of detecting oligomeric Tau at earlier ages, compared to typical ages in which neurofibrillary tangles can be detected. In terms of therapy, these oligomer-selective scFv antibodies represent an advantage over non-conformational antibodies as they do not block the physiological functions carried out by monomeric Tau.

### 3.2. Parkinson’s Disease

Parkinson’s disease (PD) is a neurodegenerative disorder associated with the abnormal aggregation of the neuronal membrane protein alpha synuclein (α-syn) (reviewed in Shulz-Schaeffer [93]). It has been shown that, besides the formation of insoluble aggregates that deposit inside neurons as inclusion bodies, termed Lewy bodies, α-syn also forms neurotoxic soluble oligomers/protofibrils [94,95]. As with Aβ and Tau, antibody fragments are beginning to emerge in the literature with selectivity for oligomeric over monomeric or fibrillar forms of α-syn. Emadi and colleagues have identified two scFv antibodies of particular use in elucidating α-syn oligomer structure–function relationships. The scFv D5 was found to be selective for oligomers more abundant in initial stages of α-syn aggregation and to block further aggregation of these oligomers and their toxicity in SH-SY5Y cells [76]. D5 was also seen to interact with oligomers formed by the Huntington’s disease-associated protein htt51Q [96], in line with the notion that many antibodies raised against amyloid oligomers cross-react with structurally similar oligomers formed by non-related proteins [97]. In contrast, 10H, an scFv that targets oligomers more abundant in later stages of α-syn aggregation, appears to be selective for oligomers of α-syn [77]. Both scFvs D5 and 10H provided neuroprotection in an α-syn overexpressing transgenic mouse model when fused to penetratin (a cell-penetrating peptide), raising a potential immunotherapeutic benefit of these scFvs in PD [98]. Although in principle antibodies targeting pan-amyloid aggregates such as scFv D5 may represent a promising therapeutic strategy, it is also important to consider that cross-reactivity may be harmful in some cases. For instance, Kvam et al. (2009) showed that the anti-fibrillar α-syn scFv-6E, which also binds mutant huntingtin and ataxin-3, increased the aggregation of these polyglutamine-rich proteins in striatal cells, aggravating intracellular dysfunction and cell death [99].

Although few antibody fragments selective for oligomeric α-syn conformations have been reported in the literature, studies utilizing antibody fragments selective for linear α-syn sequences (i.e., non-conformational antibodies) have increased our understanding of α-syn aggregation and toxicity. Zhou et al. (2004) reported the scFv antibody D10, which presented nanomolar affinity for α-syn monomers and inhibited aggregation to oligomeric and protofibrillar forms. The authors localized the D10 epitope within the C-terminus of α-syn, suggesting that perturbation in this region interferes with the aggregation process. In the same study, it was also shown that co-expression of D10 in HEK293 cells engineered to overexpresses α-syn reduced the formation of high-molecular weight α-syn aggregates, thus suggesting a positive action of D10 as an intrabody [100] (i.e., a fragment antibody engineered to accumulate within its producing cell). The VHH single domain antibodies NbSyn2 and NbSyn87 have been used to identify the role of different C-terminal regions of α-syn in fibril formation [101,102,103]. NbSyn2, which recognizes an epitope between residues 136–140, did not affect fibril formation [78,102,103]. In contrast, NbSyn87, which recognizes an epitope comprised by residues 118–128, induced conformational changes on both secondary and tertiary structures of α-syn, consequentially reducing the half-time of fibril formation [78,101].

scFvs targeting the α-syn nonamyloid component (NAC) have also shown therapeutic promise in pre-clinical studies. The NAC presents a high tendency to adopt β-pleated sheet structures and is known to play a key role in the aggregation and toxicity of α-syn in vitro and in vivo [104]. In 2008, Lynch and colleagues showed a novel NAC-selective scFv named NAC32 capable of reducing the aggregation and neurotoxicity of α-syn aggregates [105]. Other single domain antibodies targeting the NAC, NAC1 and VH14, acted similarly to NAC32 in preventing a-syn aggregation [106].

Although considerable advances towards the understanding of α-syn aggregation and toxicity have been attained by the use of fragment antibodies, few reports have been published so far evaluating the consequences of the in vivo expression/administration of these antibody fragments. Although few, these reports do demonstrate therapeutic promise. In one of these studies, the single-domain antibodies VH14 and NbSyn87 were expressed in fusion with the proteasome-targeting PEST motif, resulting in increased cytoplasmic solubility and enhanced degradation of α-syn in neuronal cell lines [78]. In another interesting piece of work, Spencer et al. (2014) induced the expression of a scFv directed to α-syn oligomers in fusion with the low-density lipoprotein receptor-binding domain from apolipoprotein B (LDL ApoB) in vivo [79]. This construction increased the penetration of the scFv into the brain via the endosomal sorting complex required for transport (ESCRT) pathway, consequently leading to lysosomal degradation of α-syn aggregates [79]. These exciting reports suggest the feasibility of in vivo expression of engineered anti-oligomeric scFvs as a therapeutic alternative for PD.

### 3.3. Huntington’s Disease

Huntingtin (HTT) is a ubiquitously expressed large protein (3144 amino acids) involved in the pathogenesis of Huntington’s disease (HD) [107]. Although the diverse physiological roles of HTT are not yet fully understood, it is well known that its aggregation and neurotoxicity are dependent on the presence of an aberrant polyglutamine (polyQ) stretch encoded in exon 1 of the htt gene (corresponding to the N-terminus in the protein) [108,109,110]. In mutant-disease-associated HTT, this polyQ stretch is longer than in wild type HTT, reaching 40 or more glutamine residues (as opposed to normally 20 on average) [110]. Interestingly, this increment is enough to impact the stability of the whole molecule, driving its aggregation into both soluble oligomers and insoluble aggregates [111].

Since HTT aggregates are exclusively intraneuronal, intrabodies have been the antibody fragment type preferentially applied to their structure-function study and their therapeutic targeting. One of the first scFv-type intrabodies directed to huntingtin was reported by Lecerf et al. (2001). Named C4, this scFv binds to residues 1–17 of HTT, a sequence N-terminal to the polyQ repeat in HTT exon 1, stabilizing an alpha helix-rich oligomeric complex and preventing amyloid formation [112,113]. When co-expressed with HTT exon 1 in non-neuronal cells, C4 was capable of reducing the amount of HTT aggregates and redirecting the subcellular localization of HTT exon 1. Moreover, C4 efficiently reduced cell death in malonate-treated brain slice cultures expressing mutant HTT [114]. Additionally of importance, expression of C4 in the HD disease mouse model B6.HDR6/1, via AAV2/1 vector, led to delayed HTT aggregation in both early and late disease stages [115]. The authors also generated scFv-C4 in fusion with the PEST domain to increase proteasomal degradation of the antigen–antibody complex [115].

A piece of pioneering work by Khoshnan et al. (2002) reported three scFvs (MW1, MW2, and MW7) produced by cloning the antigen-binding domains of monoclonal IgGs targeting either polyQ or an adjacent domain in HTT exon 1 rich in proline residues (named PRD) into scFv scaffolds [116]. The scFv MW7, selective for PRD, inhibited cell death induced by mutant HTT in co-transfected HEK293 cells [116]. Surprisingly MW1 and MW2, both selective for polyQ, accelerated aggregation and cell death in the same culture model. Possible explanations for this unexpected result are that MW1 and MW2 either stabilized a toxic aggregated conformation of HTT or interfered with the binding of HTT to other molecules mediating HTT toxicity [116]. These findings highlight the complexity and importance of identifying fragment antibodies that indeed target toxic oligomeric species, which are expected to show promise as therapeutics and/or diagnostics.

In another piece of work, multiple intrabodies targeting HTT PRD domains (scFv MW7; VL Happ1; VL Happ3) or the HTT N-terminus (VL 12.3) were used to investigate the role of these domains in HTT aggregation and toxicity [117]. VL 12.3 had been previously shown to reduce toxicity in a neuronal culture model of HD [118]. All of these intrabodies reduced mutant HTT exon 1 aggregation and toxicity in both cell culture and brain slice models of HD, although the mechanisms of protection were different. While the N-terminus-targeting intrabody altered HTT subcellular localization, the PRD-targeting intrabodies were seen to increase the turnover rate of HTT [119]. These results reinforce the notion of a strong correlation between the structural domains targeted by each intrabody and their mechanism of neuroprotection. Fragment antibodies VL 12.3 and Haap1 were also employed to investigate the contribution of N-terminus and PRD domains to HD pathology in vivo using five different HD mouse models. While VL 12.3 showed no significant effects on one model, and increased mortality in another, Haap1 alleviated HD neuropathology in all the five animal models tested, including prolonged lifespan in one model [120].

Finally, the scFv-EM48, which targets the C-terminus of human mutant HTT exon 1, also showed promising results in an HD mouse model, as decreased formation of neuropil aggregates and cognitive HD-like symptoms [114]. In conjunction with data obtained with antibody fragments targeting the N-terminus, the polyQ domain, and the PRD domain, these data indicate that all domains within HTT exon 1 play a role in mutant HTT aggregation and toxicity. When used as an intrabody, scFv-EM48 also suppressed the toxicity of mutant HTT in HEK293 cells. The ability of this antibody fragment to increase the ubiquitination and consequent degradation of cytoplasmatic HTT suggests that scFv-EM48 acts by promoting the cytoplasmic clearance of mutant HTT thereby preventing its accumulation.

### 3.4. Prion Diseases

Prion diseases are characterized by the brain accumulation of aggregated and neurotoxic forms of the prion protein (PrP). Under physiological conditions, PrP presents as a ~24 kDa transmembrane protein that exerts a number of functions, such as metal ion hemostasis and cell adhesion [121]. On the other hand, in diseased brains, it converts into a beta-sheet-rich confirmation named PrP^sc^ (i.e., the scrapie isoform), which forms both soluble oligomers and amyloid fibrils [122,123,124]. Importantly, PrP^sc^ is known to catalyze the conversion of harmless PrP molecules into the aggregation-prone conformation PrP^sc^, thus conferring to Prion diseases their unique infectious nature [123]. Finding molecules capable of inhibiting either the formation or the toxicity of PrP^sc^ aggregates, including soluble oligomers, has been a major goal in the prion diseases field, as a way to provide a disease-modifying therapy for patients. In this regard, some fragment antibodies have been selected that display promising inhibitory activity on PrP^sc^ oligomerization and fibrillization both in vitro and in cellular models [125,126].

In 2001, Peretz et al. reported the Fab antibody fragment D18, which binds to an epitope within residues 132–156 in helix 1 of the Prion protein in its native conformation, a region thought to contribute to PrP^sc^ assembly and prion elongation. Although the aggregation states targeted by D18 have not yet been identified experimentally, D18 was found to inhibit prion elongation in cultured mouse neuroblastoma cells infected with PrP^sc^ [126]. Subsequently, Campana et al. (2009) engineered scFv-D18 from Fab-D18 and used in silico tools to create a structural model of scFv-D18 bound to PrP. In that model, PrP residue Arg151 was seen to be key in the interaction with the antibody fragment, by anchoring PrP to the cavity formed on antigen binding site of the scFv [127].

More recently, Fujita et al. (2011) cloned the variable region of mAb 3S9—previously shown to inhibit PrP^sc^ accumulation in cell lines infected with mouse-adapted scrapie strains [128,129]—into the scaffold of a scFv antibody. The resulting antibody, named scFv-3S9, recognized an epitope containing Tyr154 in the helix 1 of PrP. When injected into mice brains, Prion-infected cells expressing scFv-3S9 presented less Prion pathology than infected cells not expressing this scFv [128].

Lastly, Sonati and coworkers (2013) used a panel composed of full-length antibodies and antibody fragments (Fab and scFv) directed to either the globular domain or the flexible tail on PrP, to investigate the role of these regions in oligomerization and neurotoxicity. Results generated on cerebellar organotypic cultured slices showed that both domains are required for toxicity, as the flexible tail acquires oxidative stress-mediated toxicity upon undergoing a conformational change originated from the globular domain [130]. This comprehensive work reinforced the notion that antibody-based therapeutic developments against Prion diseases must include a detailed analysis of the targeted structural epitope of each antibody candidate as well as the molecular and clinical outcomes of targeting these epitopes.

## 4. Concluding Remarks

Increased knowledge about the aggregation pathways and conformations of the toxic aggregate species relevant to CNS amyloidoses has been obtained with the use of fragment antibodies, in particular Fab, scFv, and VHH (Table 1). As technologies for engineering fragment antibodies are constantly improving, the perspective for the generation of novel fragment antibodies with high selectivity for toxic oligomeric conformations as diagnostic and/or therapeutic candidates for CNS amyloidoses, is also rising.

Methodologies for rational Aβ-targeting antibody design have been reviewed (e.g., Plotkin and Cashman, 2020 [131]). For example, just as our group has successfully generated full-length IgGs with selectivity for AβOs over monomers and fibrils [132], rational immunization with specific toxic AβO species can be employed, followed by conversion of the resulting anti-AβO IgG to an antibody fragment. Alternatively, specific toxic AβO species can be utilized in rational bio-panning of antibody fragment libraries. These specific AβO species can be generated by size-based separation methods (reviewed in [12]) or by utilizing specific Aβ monomeric proteoforms ([133,134]) and can be stabilized by various methods. For example, chemical crosslinking via DFDNB (1,5-difluoro-2,4-dinitrobenzene) has been shown to stabilize high molecular weight AβOs that exhibit toxicity in cell cultures and in vivo [135]. Alternatively, computational prediction of regions present on the surface of toxic oligomeric species is emerging as an additional strategy for rational identification of target species [136].

We envision that the use of fragment antibodies in structural studies aimed to unravel the molecular mechanisms of protein aggregation and related toxicity has a strong potential to make unique contributions to the field. In conjunction with CDR mapping and the detailed analysis of the assembly selectivity of each fragment antibody described, this approach may significantly improve our knowledge regarding key atomic contacts between antibodies and toxic oligomers, and as a consequence, the structural moieties that confer toxicity to amyloid oligomers. These advances could enhance the field’s capability of engineering antibody fragments able to selectively target neurotoxic aggregates amongst a multitude of oligomeric assemblies co-existing in diseased human tissue. Even in a likely case in which different oligomeric species contribute to neurotoxicity, and thus a single, highly specific antibody would not able to fully neutralize the pathogenic cascade, a therapeutic strategy based on the combination of multiple oligomer-selective antibody fragments directed against different species could be employed to circumvent this issue.

The cognitive benefit and lowering of multiple AD markers reported in AD patients treated with the antibody aducanumab (Biogen)—a monoclonal IgG that preferentially targets aggregated Aβ [137,138]—has brought hope, reinforcing the notion that selectively targeting neurotoxic aggregates would guide the field toward disease-modifying treatments against brain amyloidosis. Indeed, the FDA has recently granted aducanumab priority review [139]. However, there is still room for improvement in the field as the therapeutic benefits of aducanumab were only apparent following a re-analysis of the phase three trials that were initially halted due to a lack of efficacy [138]. In our view, this improvement will stem from the development of antibodies even more selective to neurotoxic oligomeric assemblies. In this context, detailed structural information on these toxic oligomers will be invaluable to the targeted design of new oligomer-selective fragment antibodies with improved specificity and clinical outcomes.

## Figures and Tables

**Figure 1 ijms-21-08920-f001:**
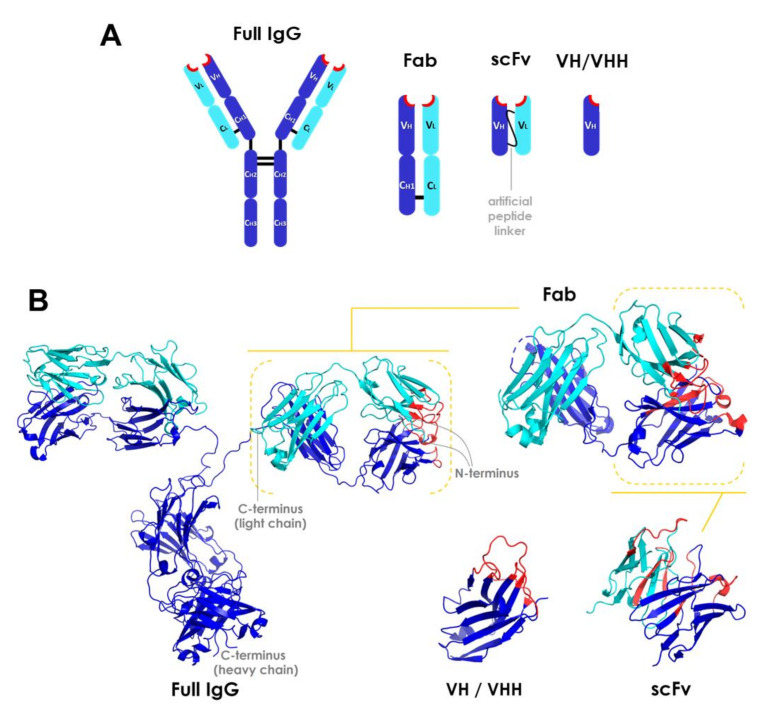
Overview of the structure of antibody fragments. (**A**) General schematic of domain framework and (**B**) ribbon diagrams of full-length IgG and fragment molecules. Structures were obtained from the Protein Data Bank (http://www.rcsb.org/pdb/). CH, CL, VH, and VL stand for constant heavy, constant light, variable heavy, and variable light domains, respectively. Heavy or light chains are depicted in dark blue or cyan, respectively. Complementarity-determining region (CDR) segments are highlighted in red. PDB codes: 1IGT (full length IgG), 5VH3 (Fab), 4NKO (scFv), and 3R0M (VHH) [58]. Fab: fragment antigen-binding; scFv: single chain variable fragments; VH/VHH: heavy chain variable domain fragment.

**Figure 2 ijms-21-08920-f002:**
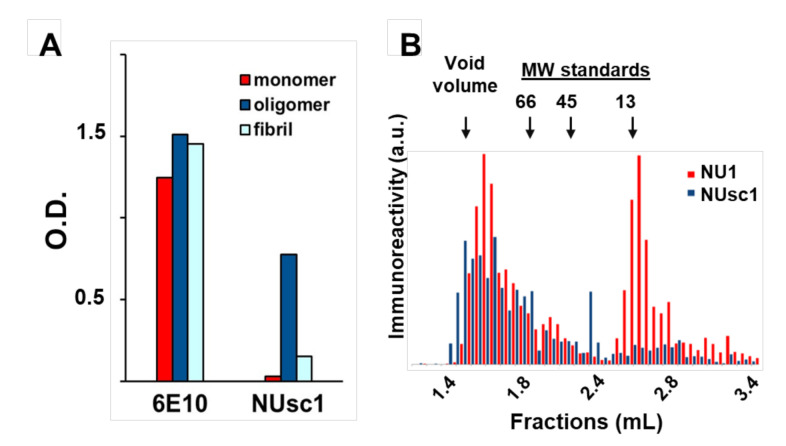
The scFv antibody NUsc1 is highly selective to high molecular weight Aβ oligomers (AβO). (**A**) NUsc1 shows high selectivity for Aβ oligomers over monomers and fibrils as determined via ELISA. The anti-pan Aβ IgG 6E10 is shown for comparison. Adapted with permission from (Velasco et al., ACS Chem. Neurosci. 2012 [64]). Copyright (2020) American Chemical Society. (**B**) Within a synthetic AβO population, NUsc1 selectively targets a high molecular weight subset, showing little binding to a lower molecular weight subset that is readily bound by the anti-AβO IgG NU1. Reactivity of both antibodies to AβO fractions separated by size-exclusion chromatography under non-denaturing conditions was determined by dot immunoblotting. Reprinted with permission from (Sebollela et al., Journal of Neurochem. 2017 [65]). Copyright (2020) John Wiley and Sons.

**Table 1 ijms-21-08920-t001:** Conformation-sensitive antibody fragments directed to oligomeric species of proteins implicated in central nervous system (CNS) amyloidoses.

Antibody	Fragment Type	CNS Amyloidosis	Target		Ref.
			High Affinity *	Low Affinity	
NUsc1	scFv	AD	Aβ42 Oligomers (>50 kDa) ^¶^	not reported	[64,65]
MO6	scFv	AD	Aβ42 Oligomers and Immature fibrils (18–37 kDa #)	not reported	[66]
AS	scFv	AD	Aβ42 Oligomers and Immature Protofibrils (25–55 kDa #)	not reported	[67]
HT6	scFv	AD	Aβ42 Oligomers (18–45 kDa ^#^)	not reported	[68]
11A5	scFv	AD	Aβ42 Oligomers (34 kDa ^#^)	not reported	[69]
A4	scFv	AD	Aβ42 Oligomers	Aβ42 Monomers and Fibrils	[70]
E1	scFv	AD	Aβ42 Oligomers	not reported	[71]
scFv59	scFv	AD	Aβ42 Oligomers and Plaques	not reported	[72]
scFv235	scFv	AD	phosphoTau Oligomers (50–70 kDa) #	Tau monomers	[73]
F9T, D11C, H2A	scFv	AD	Tau Oligomers (Trimers) ^¶^	not reported	[74]
RN2N	scFv	AD	Tau Oligomers	not reported	[75]
D5	scFv	PD	α-Synuclein Oligomers	not reported	[76]
10H	scFv	PD	α-Synuclein Oligomers (Trimers and Hexamers) ^¶^	α-Synuclein Monomers	[77]
VH14, NbSyn87	VH	PD	α-Synuclein Oligomers	not reported	[78]
D5-apoB	scFv	PD	α-Synuclein Oligomers (28–80 kDa) ^#^	not reported	[79]
W20	scFv	Various diseases	Oligomers of Aβ40 and Aβ42, PrP^C^, α-Syn, amylin, insulin, lysozyme	not reported	[80]

* MW/size of targeted oligomers is presented when available. It is also indicated whether MW/size have been determined under non-denaturing ^¶^ or denaturing ^#^ conditions. AD: Alzheimer’s Disease; PD: Parkinson’s Disease; PrP^c^: cellular prion protein.

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
