# Peer review of "Antibody Fragments as Tools for Elucidating Structure-Toxicity Relationships and for Diagnostic/Therapeutic Targeting of Neurotoxic Amyloid Oligomers"

_ijms, 2020, doi:10.3390/ijms21238920_

Round 1
Reviewer 1 Report
In this review article, Bitencourt et al. summarize antibody fragments targeting neurotoxic amyloid oligomers. Many readers will be interested in this article because these antibody fragments are useful in diagnosing and treating diseases. Overall, the article is well written, and I recommend it to be published in the International Journal of Molecular Science. However, the properties of each antibody fragment shown in this paper are only listed in the text, which may be difficult for the reader to understand immediately. Although some information is summarized in Table 1, I strongly recommend adding some tables or figures with more detailed properties of antibody fragments.
Minor comment
Line 98: Explain the abbreviations (scFv and VH/VHH). There are explanations for them below (line 103 and line 107), so move them to the first part (line 98). I think these abbreviations in Abstract also need explanation.
Author Response
Reviewer #1
“In this review article, Bitencourt et al. summarize antibody fragments targeting neurotoxic amyloid oligomers. Many readers will be interested in this article because these antibody fragments are useful in diagnosing and treating diseases. Overall, the article is well written, and I recommend it to be published in the International Journal of Molecular Science. However, the properties of each antibody fragment shown in this paper are only listed in the text, which may be difficult for the reader to understand immediately. Although some information is summarized in Table 1, I strongly recommend adding some tables or figures with more detailed properties of antibody fragments.”
We thank the Reviewer for the positive assessment of our manuscript. In the revised version, we have expanded the Table I, which now provides additional information on the target selectivity of the antibody fragments.
“Line 98: Explain the abbreviations (scFv and VH/VHH). There are explanations for them below (line 103 and line 107), so move them to the first part (line 98). I think these abbreviations in Abstract also need explanation.”
In the revised manuscript, we have moved explanations for abbreviations to the first appearance of these terms (now line 100). As requested, we have also explained these terms in the abstract.
Please see the attachment.

Reviewer 2 Report
This manuscript presents a detailed summary of recent literature related to antibody fragments selective for oligomers that are associated with different CNS amyloid diseases. The manuscript highlights advantages and disadvantages of antibody fragments and gives a number of case examples where this methodology has shown promise for binding toxic oligomers and inhibited cell-toxicity based on amyloid formation.
Despite the well-organised review of the literature, I felt that the authors have not really given their own insights/commentary about this topic in a way that provokes a reader to think more critically of the literature presented, and this is a vital part of writing review articles. Therefore, I suggest that the authors expand their summary/concluding statement with some of their opinion/insights. Some areas of thought that were raised as I read this manuscript include the points below. Although I do not expect all of these to be addressed, some commentary to this effect needs to be included. I would be happy for the authors to raise their own perspectives but felt it would be useful to give some potential examples.
- In the conclusion, the authors state that improved knowledge regarding key atomic contacts will inform on the structural moieties that confer toxicity to amyloid oligomers; can the authors provide some insights on the advantages/disadvantages of the development of antibody fragment for the conformation of oligomers (i.e. like Kayed/Glabe OC antibody) or is the future based on protein specific oligomer binders?
- Given the recent focus on monoclonal antibodies like aducanumab, ganternerumab etc., can the authors offer some commentary on how antibody fragments could address some of the issues of these potential therapeutics?
- Can the authors offer some specific ways that oligomer binders can be designed? i.e. there are a number of stable oligomer systems (and many show cellular toxicity), can designs be made based on these? Rational designing methodologies? Etc.
- I think that with oligomer-binding biomolecule, there is always a potential for multiple-binding species and so a critical analysis of how this could be advantageous (or indeed, deleterious) to the amyloid cascade would be appropriate.
Points to address:
1) In reference to the synuclein-based Nb87 nanobody; the authors have not really addressed the fact that this particular nanobody, although has a mode of action of binding to, and preventing the toxicity of, oligomers, has a 14.2 nM Kd for monomeric synuclein (Guilliams et al (2013) JBC). Also, in Iljina et al. (2017) it is noted that the inhibition studies were done with a 2-fold excess of the nanobody (in relation to alpha-synuclein). Therefore, the binding of Nb87 to monomer in the action of inhibition cannot be ignored and an appropriate expansion of this needs to be included in this example (perhaps in others?).
2) In the Table 1, please could they authors make sure to indicate all other species that the antibody fragments may bind to? I think this table over simplifies the complicated nature of finding binders to oligomer species.
3) Some of your references are missing details
Author Response
Reviewer #2:
“This manuscript presents a detailed summary of recent literature related to antibody fragments selective for oligomers that are associated with different CNS amyloid diseases. The manuscript highlights advantages and disadvantages of antibody fragments and gives a number of case examples where this methodology has shown promise for binding toxic oligomers and inhibited cell-toxicity based on amyloid formation.
Despite the well-organised review of the literature, I felt that the authors have not really given their own insights/commentary about this topic in a way that provokes a reader to think more critically of the literature presented, and this is a vital part of writing review articles. Therefore, I suggest that the authors expand their summary/concluding statement with some of their opinion/insights. Some areas of thought that were raised as I read this manuscript include the points below. Although I do not expect all of these to be addressed, some commentary to this effect needs to be included. I would be happy for the authors to raise their own perspectives but felt it would be useful to give some potential examples.”
We thank the Reviewer for the positive assessment of our manuscript, and for the constructive criticism. To provide to the readers more information on our view of the field, as suggested, in the revised manuscript we have added a new Figure (Fig. 2), in which we present a representative piece of our published work using an interesting antibody fragment showing a striking size-selectivity for a subpopulation of Aβ oligomers. Also following the Reviewer’s recommendation, we have included our thoughts on all the points raised by the Reviewer, as detailed below.
- “In the conclusion, the authors state that improved knowledge regarding key atomic contacts will inform on the structural moieties that confer toxicity to amyloid oligomers; can the authors provide some insights on the advantages/disadvantages of the development of antibody fragment for the conformation of oligomers (i.e. like Kayed/Glabe OC antibody) or is the future based on protein specific oligomer binders?”
- “I think that with oligomer-binding biomolecule, there is always a potential for multiple-binding species and so a critical analysis of how this could be advantageous (or indeed, deleterious) to the amyloid
In the revised manuscript, we have expanded a paragraph in the “Concluding remarks” section to cover both of these points. The revised passage (now lines 420-431) is below.
“We envision that the use of fragment antibodies in structural studies aimed to unravel molecular mechanisms of protein aggregation and related toxicity has a strong potential to make unique contributions to the field. In conjunction with CDR mapping and the detailed analysis of the assembly selectivity of each fragment antibody described, this approach may significantly improve our knowledge regarding key atomic contacts between antibodies and toxic oligomers, and as a consequence, the structural moieties that confer toxicity to amyloid oligomers. Those advances could enhance the field’s capability of engineering antibody fragments able to selectively target neurotoxic aggregates amongst a multitude of oligomeric assemblies co-existing in diseased human tissue. Even in a likely case in which different oligomeric species contribute to neurotoxicity, and thus a single, highly specific antibody would not able to fully neutralize the pathogenic cascade, a therapeutic strategy based on the combination of multiple oligomer-selective antibody fragments directed against different species could be employed to circumvent this issue. “
- “Given the recent focus on monoclonal antibodies like aducanumab, ganternerumab etc., can the authors offer some commentary on how antibody fragments could address some of the issues of these potential therapeutics?”
In the revised manuscript we have added the following commentary on this regard in (Concluding remarks, line 432)
“The cognitive benefit and lowering of multiple AD markers reported in AD patients treated with the antibody aducanumab (Biogen), a monoclonal IgG that preferentially targets aggregated Aβ [138,139], has brought hope, reinforcing the notion that selectively targeting neurotoxic aggregates would guide the field toward disease-modifying treatments against brain amyloidosis. Indeed, the FDA has recently granted aducanumab priority review [140]. However, there is still room for improvement in the field as the therapeutic benefits of aducanumab were only apparent following a re-analysis of the Phase 3 trials that were initially halted due to lack of efficacy [139]. In our view, this improvement will stem from the development of antibodies even more selective to neurotoxic oligomeric assemblies. In this context, detailed structural information on these toxic oligomers will be invaluable to the targeted design of new oligomer-selective fragment antibodies with improved specificity and clinical outcomes.”
- “Can the authors offer some specific ways that oligomer binders can be designed? i.e. there are a number of stable oligomer systems (and many show cellular toxicity), can designs be made based on these? Rational designing methodologies? Etc.”
In the revised manuscript we have added the following commentary on this regard in (Concluding remarks, line 408)
“Methodologies for rational Aβ-targeting antibody design have been reviewed (e.g., Plotkin and Cashman, 2020 [132]). For example, just as our group has successfully generated full-length IgGs with selectivity for AβOs over monomers and fibrils [133], rational immunization with specific toxic AβO species can be employed, followed by conversion of the resulting anti-AβO IgG to an antibody fragment. Alternatively, specific toxic AβO species can be utilized in rational bio-panning of antibody fragment libraries. These specific AβO species can be generated by size-based separation methods (reviewed in [12]) or by utilizing specific Aβ monomeric proteoforms ([134,135]) and can be stabilized by various methods. For example, chemical crosslinking via DFDNB (1,5-difluoro-2,4-dinitrobenzene) has been shown to stabilize high molecular weight AβOs that exhibit toxicity in cell cultures and in vivo [136]. Alternatively, computational prediction of regions present on the surface of toxic oligomeric species is emerging as an additional strategy for rational identification of target species [137].”
Please see the attachment.

Reviewer 3 Report
The authors presented an interesting and well-written review of the topic. They carefully collected and discussed the results of recent papers dedicated to the usage of antibodies against protein aggregates. And I suppose that the review will be useful for the specialists in this field.
Minor comments.
In abbreviations, PrPC and PrPSc "C" and "Sc" should be written in superscript.
Lines 90 and 198 "in vitro" should be italicized.
Table 1. I would suggest using the abbreviations of Alzheimer’s and Parkinson's disease, AD, and PD, respectively, to make the table more readable. Also, I was surprised to see the gene name SNCA among the antibody targets.
Line 408 "htt" in Abbreviations should be replaced by "HTT", used in the text.
The abbreviation of PrPSc should be added to the abbreviation list. Also, I would suggest rechecking the abbreviation list, maybe several abbreviations can be removed from it. For example, BBB was used only once in the text.
Finally, I would suggest removing the section Acknowledgments, which seems to come from the template without modifications.
Author Response
Reviewer #3:
The authors presented an interesting and well-written review of the topic. They carefully collected and discussed the results of recent papers dedicated to the usage of antibodies against protein aggregates. And I suppose that the review will be useful for the specialists in this field.
We thank the Reviewer for the positive assessment of our manuscript.
- “In abbreviations, PrPC and PrPSc "C" and "Sc" should be written in superscript.”
All the errors in superscript formatting have been corrected. Thank you.
- “Lines 90 and 198 "in vitro" should be italicized.”
All the errors in italicized formatting have been corrected, including others not indicated by the Reviewer. Thank you.
- “Table 1. I would suggest using the abbreviations of Alzheimer’s and Parkinson's disease, AD, and PD, respectively, to make the table more readable. Also, I was surprised to see the gene name SNCA among the antibody targets.”
In the revised manuscript we have included a revised version of Table I, which now provides additional information on the target selectivity of the antibody fragments and also is now formatted according to Reviewer’s recommendations.
- “Line 408 "htt" in Abbreviations should be replaced by "HTT", used in the text.”
This error has been corrected. Thank you.
- “The abbreviation of PrPSc should be added to the abbreviation list. Also, I would suggest rechecking the abbreviation list, maybe several abbreviations can be removed from it. For example, BBB was used only once in the text.”
In the revised manuscript, we have added PrPsc in the abbreviation list. In addition, the abbreviation list has been updated, including with removal of some terms used only once throughout the text, and recommended by the Reviewer.
- “Finally, I would suggest removing the section Acknowledgments, which seems to come from the template without modifications.”
We have removed this section in the revised manuscript.
Please see the attachment.
